# A framework to efficiently describe and share reproducible DNA materials and construction protocols

Hideto Mori [1,2,3] & Nozomu Yachie [1,4]✉

DNA constructs and their annotated sequence maps have been rapidly accumulating with the advancement of DNA cloning, synthesis, and assembly methods. Such resources have also been utilized in designing and building new DNA materials. However, as commonly seen in the life sciences, no framework exists to describe reproducible DNA construction processes. Furthermore, the use of previously developed DNA materials and building protocols is usually not appropriately credited. Here, we report a framework QUEEN (framework to generate quinable and efficiently editable nucleotide sequence resources) to resolve these issues and accelerate the building of DNA. QUEEN enables the flexible design of new DNA by using existing DNA material resource files and recording its construction process in an output file (GenBank file format). A GenBank file generated by QUEEN can regenerate the process code such that it perfectly clones itself and bequeaths the same process code to its successive GenBank files, recycling its partial DNA resources. QUEEN-generated GenBank files are compatible with existing DNA repository services and software. We propose QUEEN as a solution to start significantly advancing the material and protocol sharing of DNA resources.

[1] Research Center for Advanced Science and Technology, The University of Tokyo, Tokyo 153-8904, Japan. [2] Institute for Advanced Biosciences, Keio University, Tsuruoka 997-0035, Japan. [3] Graduate School of Media and Governance, Keio University, Fujisawa, Kanagawa 252-0882, Japan. [4] School of Biomedical Engineering, Faculty of Applied Science and Faculty of Medicine, The University of British Columbia, Vancouver, BC V6T 1Z3, Canada. ✉email: nozomu.yachie@ubc.ca

Designing and building DNA are essential processes in most life science research today. The introduction of exogenous DNA into cells and animals allows for monitoring of molecular and cellular behaviors, and reverse engineering and functional enhancement of target systems. Chemical DNA synthesis[1] and assembly methods[2] have largely advanced in the last couple of decades, leading to the whole synthesis of bacterial[3,4] and yeast chromosomes[5–7] and the establishment of biofoundries towards the automated production of engineered microorganisms[8,9]. A total of several hundred thousand DNA plasmids have been deposited to public DNA repository services, such as AddGene[10] and DNASU[11]. While the growing DNA resources have been successfully accelerating life science research, previously established DNA resources have yet to be fully utilized to produce new DNA constructs efficiently. Current DNA resource sharing and building methods have major room for improvement in optimal recycling and crediting of DNA resources and protocols.

Standardization of materials and knowledge descriptions are important to best mobilize those established previously for new material development. In synthetic biology, there have been several efforts devoted for (1) the standardization of DNA modules and (2) the standardization of functional DNA annotations. To standardize DNA modules, "DNA brick" systems have been proposed in which DNA fragments are sandwiched by a limited set of compatible restriction enzyme (RE) digestion sites such that these modular DNA parts can be reused in different DNA assemblies by ligation[12–15]. Such a system would accelerate the DNA construction process when a sufficient amount of modular parts are available. (They have yet to be widely used by the community, probably because the number of compatible parts have yet to meet the diverse demands in biology, or because PCR-based fragment preparation and highly specific overlap DNA assembly methods[16–21] have become the mainstream and freed molecular cloning from RE-based techniques.) The synthetic biology open language (SBOL)[22] has been proposed to provide functional annotation of gene circuits encoded in DNA sequences. SBOL enables users to define genetic parts and their functional wiring diagrams like electronic circuits in a computer readable format, consequently allowing the design of sequences that confer specific functions through simulating different circuits.

While the DNA brick systems and SBOL have contributed to the efficient collection of highly modular DNA resources and the systematic annotation of functional DNA resources, respectively, they focus on the standardization of input and output DNA materials. The third key angle to advance new DNA construction with rapidly accumulating DNA resources would be the development of a standardized "process ontology" or "protocol language" to describe DNA construction processes. We envision an efficient DNA design and construction system, whereby the most optimal construction protocol of a target DNA is autonomously formulated with a combination of maximal use of existing DNA materials and knowledge on how previous DNA materials have been practically constructed. We are, however, one step away from establishing such a system. Most of us have still been designing DNA manually with GUI software tools, and the protocols have been described in natural language.

There are three major challenges to realize a process ontology of DNA construction that is widely accepted by the life science community. First, a standardized framework needs to be developed to universally frame any types of changes in DNA sequences with various types of annotations. While the methods to build and alter DNA materials have been diversifying, such as we are seeing with genome editing technologies[23], the current DNA editing software tools cannot assist users in incorporating new

methods or allow them to develop new plugins. Second, there needs to be an innovative way to motivate the community without substantial costs. Third, such a process ontology would require another system to evaluate the reproducibility and completeness of protocols, which also needs to be accepted by the community. Accompanying these three challenges, a system that globally traces the inheritance of previously established DNA material and protocol resources would be largely beneficial to measure the impacts of previous resources and to credit their developers.

A quine in computer science is a program that replicates a copy of itself without the need for any inputs. The intriguing concept of quine, which first appeared in the mid-twentieth century[24], provides us fruitful thoughts on self-replicating machines and their potentials. In this study, inspired by this concept, we developed a simple, versatile framework called QUEEN (a framework to generate "quinable" and efficiently editable nucleotide sequence resources) that resolves all the above-mentioned challenges and enables efficient description, sharing, and crediting of DNA materials and building protocols[25]. QUEEN is a Python package where step-by-step building processes of DNA and their dynamic sequence changes can be freely described and simulated. QUEEN-generated DNA products can be output in the GenBank (gbk) file format that is widely used to describe annotated DNA maps. Due to its flexible file format, QUEEN-generated gbk files achieve several unique features. First, they can regenerate the quine codes that generated themselves using QUEEN (Fig. 1a). Just like the research community has already been doing, DNA materials can be deposited to a DNA repository service together with their gbk files created by QUEEN (Fig. 1b). While it serves as a regular gbk file, the file's construction process information can also be recovered by QUEEN, where the reproducibility of the protocol is certified by this clonability. Second, in designing a new DNA construct using QUEEN, when DNA parts are taken from existing QUEEN-generated gbk files (Fig. 1c) or a protocol generated from a QUEEN-generated gbk file is modified (Fig. 1d), the newly produced gbk file inherits the parental material information as well as their building process histories. This feature will be a backbone for establishing new ways of evaluating genetic resources, protocols, and developers based on how they are inherited in the community (Fig. 1e). QUEEN-generated gbk files can easily be spread through the existing databases and DNA repository services as they can be treated as regular gbk files by existing tools. This feature will prime the promotion of the new material and protocol sharing model.

## Results

**Queen.** QUEEN is a Python module. It was designed so that other software tools and databases can easily be developed to enhance QUEEN's basic capabilities. In QUEEN, double-stranded (ds) DNA objects with their annotated feature objects are first provided as inputs. A dsDNA object can be given by specifying a DNA sequence or importing a sequence file in GenBank or FASTA file format. The dsDNA objects can then be manipulated by using (i) four basic operational functions, "cut," "flip," "join," and "modify ends" and (ii) two search functions for DNA sequence and annotation features, which can collectively represent any of the standard molecular cloning processes. We also prepared (iii) two other direct edit functions for DNA sequence and annotation features (Fig. 2). A dsDNA object generated or edited by QUEEN can be output into a gbk file that encodes the quine code to replicate the entire operational process.

A linear or circular dsDNA object can be segmented into multiple fragment objects or linearized by "cut." Each cut site can be defined by a single DNA sequence position (blunt-end cut), two

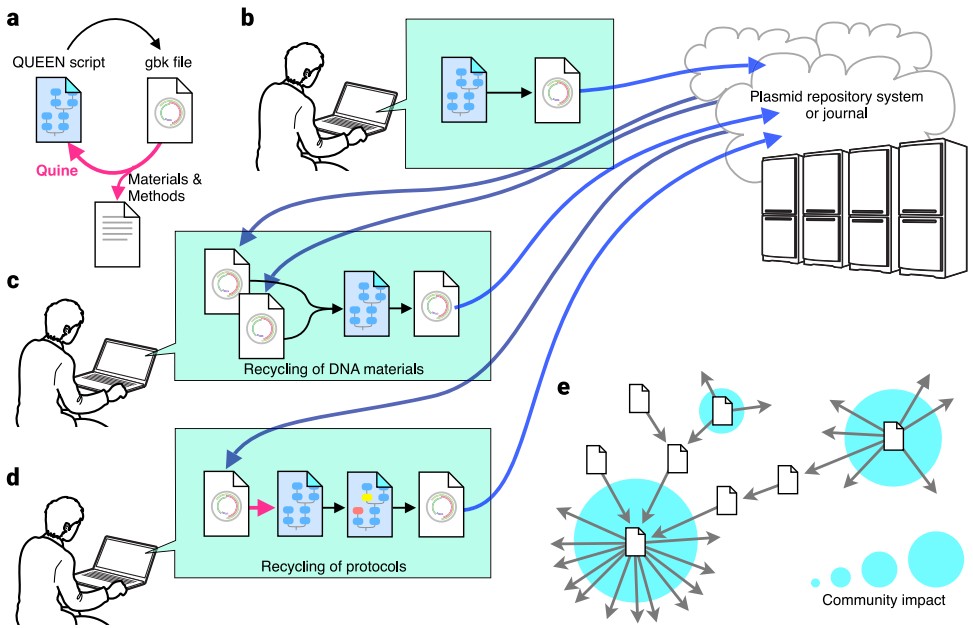

**Fig. 1 QUEEN. a** QUEEN enables the generation of a quine code from a gbk file. **b** A QUEEN-generated gbk file can be deposited to an existing DNA repository as a regular gbk file. **c** When DNA sequences are partially or fully inherited from previous QUEEN-generated gbk files to build a new DNA, the producing gbk file can contain all the production histories of the ancestral gbk files. **d** When a QUEEN script obtained from a previously established gbk file is modified to design the building process of a new DNA, such a history is also inherited by the produced gbk file. **e** A QUEEN-generated gbk file traces its previously utilized DNA materials and protocols, enabling the assessment of community impact of each DNA resource or developer.

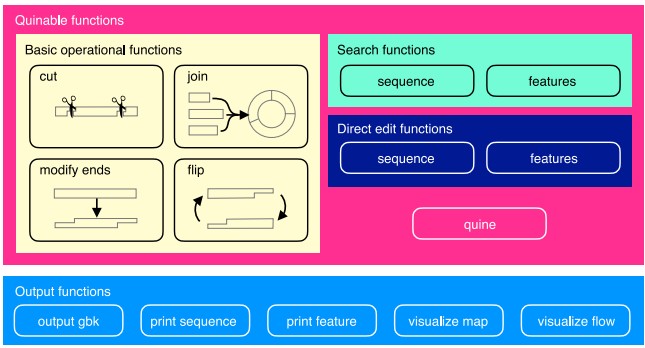

**Fig. 2** Overview of QUEEN operational functions.

sequence positions for both strands (sticky-end cut), or a feature object that represents an RE rule. Any feature objects, such as gene names and primer binding sites, are inherited from the input dsDNA objects to the corresponding sequence regions of the newly generating objects. Annotation feature objects on the cut boundaries that are split are also passed to the new fragments, each with a qualifier specifying it as a broken feature. For convenience, we also implemented "crop" as a branch function of "cut" to extract a segment bound by two cut sites. Any number of dsDNA objects can be assembled at once by "join," which requires the connecting DNA end structures to be compatible (i.e., only blunt ends and compatible sticky ends can be joined). If the assembly reconstitutes any sequences that are associated to any feature objects annotated in the parental dsDNA, the same feature objects are also restored in the generating dsDNA. Any single-stranded (ss)DNA sequence can be added to or removed from whichever strand of dsDNA ends by the "modify ends" operation. This operation enables the description of any overlap-based DNA assembly methods requiring long overlapping DNA end sequences by changing such ends to sticky ends. dsDNA fragments can also be "flipped." This operation can be used to, for example, model

DNA sequence inversion via site-specific DNA recombination (e.g., Cre-loxP and FLP-FRT) with "cut" and "join."

When a user constructs a new DNA using QUEEN, they do not need to remember the complete information of all the dsDNA objects being treated and their feature objects. The search functions allow them to obtain information on target sequence and feature objects storing a target data by text search, using regular expression or fuzzy matching. Similarly, the direct edit functions enable searching of DNA sequence and feature objects, but they also enable direct editing of the target objects obtained by the search. The direct edit function of dsDNA objects can be employed to model genome editing, and the direct edit function of feature objects can enable creation of new feature objects and removal and editing of existing feature objects. To export and display information of dsDNA objects, QUEEN also provides five output functions: "output gbk," "print sequence," "print feature," "visualize map," and "visualize flow." Especially, "visualize map" and "visualize flow" generate annotated sequence maps and operational history flow charts of dsDNA objects, respectively. These output functions support users in programming QUEEN codes with an interactive programming environment, such as Jupyter Notebook, where users can promptly check progress of their DNA construction. (The links to some examples are provided in Supplementary Table 1.)

QUEEN progressively records all operations into the "history" attribute of a feature object defined for the entire dsDNA object. This history attribute stores not only the operational history for the present DNA construction but also the past operational histories of the parental dsDNA objects whose partial fragments are inherited to the present construction. Therefore, while a QUEEN-generated gbk file can be operated by other non-QUEEN supported tools, it also has abilities to produce a quine code that self-replicates and provides the information of how previous DNA materials have been manipulated and inherited to the present DNA construct. Furthermore, while programming QUEEN codes, users can group subsets of operational flows

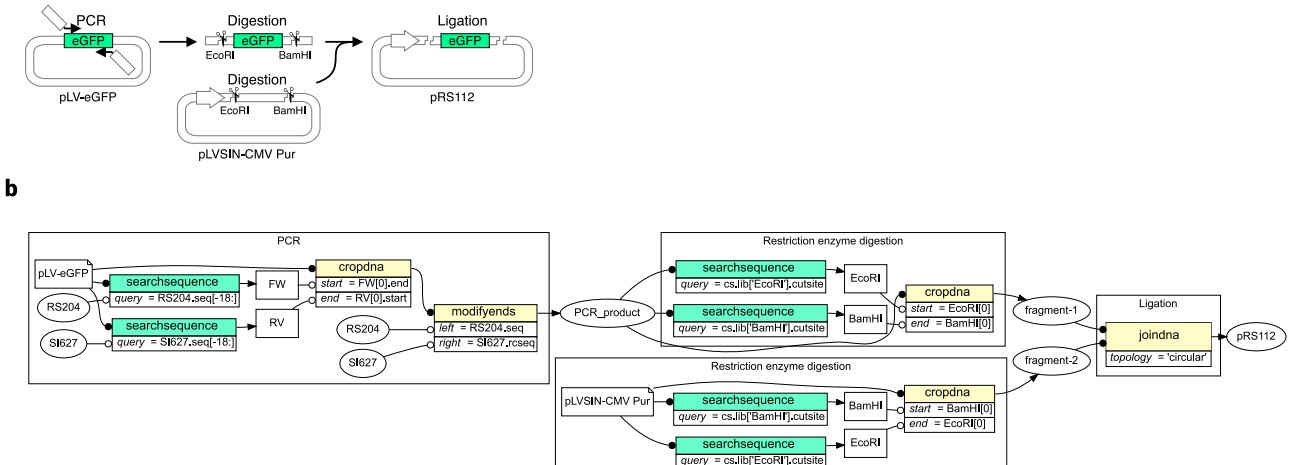

**Fig. 3 Simple molecular cloning. a** Conceptual diagram representing the cloning of eGFP into a lentiviral plasmid backbone. **b** QUEEN operational flow representing the same DNA cloning process. The visualization was directly generated from the output gbk file using QUEEN. The file shape, round, and uncolored rectangular objects represent the input gbk files, dsDNA objects, and feature objects, respectively. Colored boxes represent QUEEN operational functions with the colors corresponding to Fig. 2. Open and closed circle-headed lines represent information flows as QUEEN objects and input parameters, respectively.

and provide them with narrative descriptions of methodological procedures, which enables DNA builders to provide experimental procedures in natural language along with their process semantic descriptions. This also enables the direct generation of "Materials and Methods" descriptions from QUEEN-generated gbk files. We envision that the further development of GUI-based software tools will allow users to program the DNA construction process without making them recognize any of the above-mentioned QUEEN programming semantics.

**Simple molecular cloning**. We demonstrated that QUEEN enables the description of various DNA construction processes, simulation of dynamic DNA programs, and production of "quinable" QUEEN-generated gbk files. We first tested a description of a simple gene cloning procedure to derive a lentiviral plasmid (pRS112) that we constructed previously[26] (Fig. 3a). In this plasmid construction, an enhanced green fluorescent protein (eGFP)-encoding cassette was amplified from the pLV-eGFP plasmid by PCR with primers that have overhang sequences encoding *Eco*RI and *Bam*HI RE sites. The amplified PCR product and the destination plasmid pLV-SIN-CMV-Puro were both digested by *Eco*RI and *Bam*HI and ligated to obtain the final product.

This entire process was described using QUEEN with 11 operational steps (Fig. 3b). To obtain the PCR product, the 18-bp 3′ regions of the two primers were searched in the template. The internal DNA sequence flanked by the primer sites was then obtained by "crop," followed by the concatenation of the entire primer sequence to both ends by "modify ends" to produce the PCR product. Next, *Eco*RI and *Bam*HI cut sites and digestion patterns were searched and defined as feature objects for both the PCR product and destination lentiviral plasmid backbone. The feature objects defining the cut sites were used for the double digestion by "cut." Finally, the digested fragments with compatible sticky ends were connected by "join." We confirmed that the generated sequence was identical to that of the previously generated gbk file, and that the QUEEN-generated gbk file could produce a quine code that clones the same plasmid.

**Resource inheritance**. To demonstrate QUEEN can describe overlap DNA assembly and produce gbk files that can track their

building histories and those of inherited DNA parts, we replicated the construction processes of six CRISPR base-editor plasmids: pCMV-Target-AID, pCMV-Target-ACE, pCMV-AIDmax, pCMV-Target-ACEmax, pCMV-BE4max(C), and pCMV-ACBEmax, that we constructed previously[26] (Fig. 4a). The entire construction processes of these plasmids were based on PCR amplification of DNA fragments and Gibson Assembly, where, upon some plasmid constructions, their DNA parts were recycled for other plasmid constructions.

In brief, PCR fragment preparations were modeled by searching for primer annealing sites followed by "crop" and "modify ends," as in the preceding example above. Gibson Assembly reactions were modeled by generating long compatible sticky ends using "modify ends" and assembling them by "join." We started from four existing plasmids and one plasmid that we prepared by cloning a synthetic DNA fragment. pCMV-Target-AID, pCMV-Target-ACE, pCMV-AIDmax, and pCMV-Target-ACEmax were constructed by Gibson Assembly of fragments amplified from the initial set of plasmids, but pCMV-BE4max(C) and pCMV-ACBEmax were constructed using fragments partially obtained from pCMV-AIDmax and pCMV-Target-ACEmax, respectively. Through these assemblies, we sometimes amplified adjacent small DNA blocks for annotated DNA units separately and assembled them back in the same order in a destination plasmid for better PCR amplification of shorter fragments, rather than facing difficulties in amplifying longer PCR products. We demonstrated that even if DNA sequence regions with feature objects were once broken by this operation, they could be restored in the final plasmids (Fig. 4b and Supplementary Table 2).

We confirmed that the QUEEN scripts could generate target DNA sequences identical to those constructed previously. Their quine codes and complete operational histories could also be produced from the generated gbk files (Supplementary Figs. 1 and 2). Notably, we confirmed that the process histories of pCMV-BE4max(C) and pCMV-ACBEmax successfully inherited those of pCMV-Target-AIDmax and pCMV-Target-ACEmax, respectively (Supplementary Fig. 2). Furthermore, the construction processes of pCMV-Target-ACEmax and pCMV-ACBEmax were also able to be designed similarly to those of pCMV-Target-AIDmax and pCMV-BE4max(C), where two and three fragments were shared, respectively. Therefore, we also generated two other gbk files for pCMV-

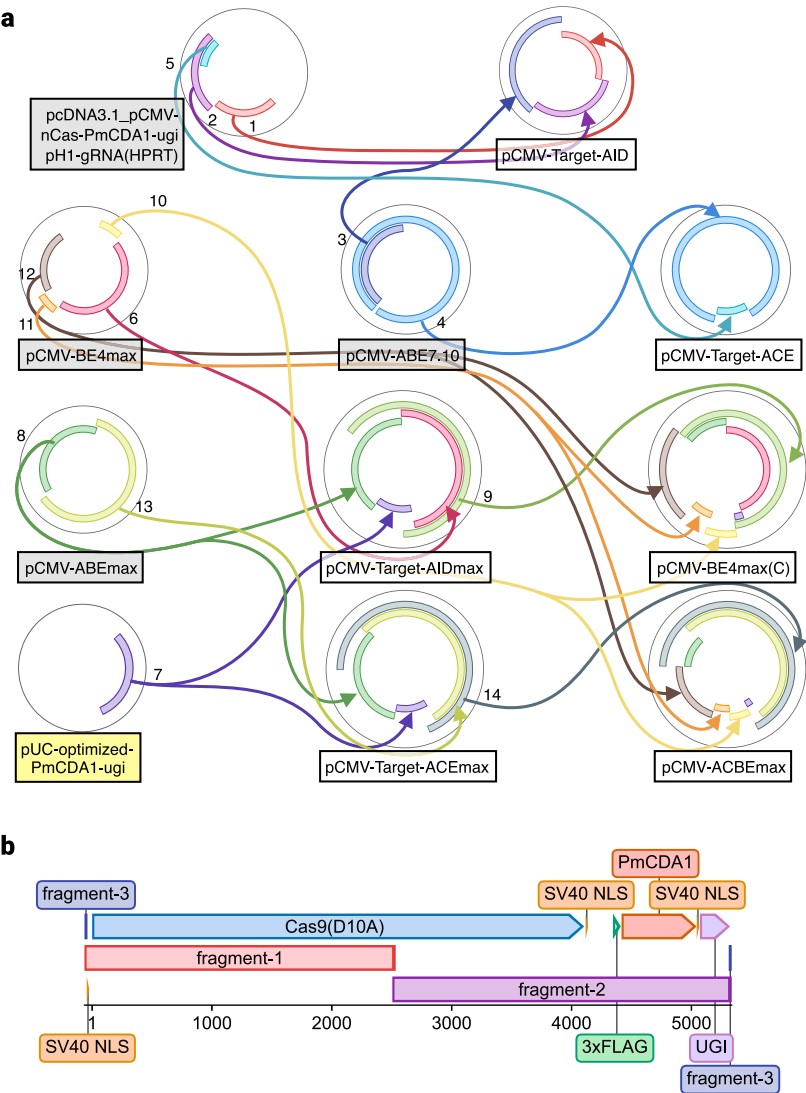

**Fig. 4 Base editor plasmid constructions. a** The construction lineages of base editor plasmids with recycling of their DNA sequences. The plasmid names with gray, yellow, and white boxes represent those obtained from Addgene, newly synthesized, or described by QUEEN in this study, respectively. **b** Recovery of the "Cas9(D10A)" feature after joining two fragments each from a different DNA object having the same feature.

Target-ACEmax and pCMV-ACBEmax by retrieving the quine codes from the pCMV-Target-AIDmax and pCMV-BE4max(C) gbk files and directly editing them. The gbk files generated through this editing strategy successfully recorded the partial inheritance of the previous protocols (Supplementary Fig. 3), showing that QUEEN enables tracking of the partial and full inheritance of both DNA materials and their construction processes.

**Simulation of dynamic DNA changes**. As seen in yeast mating-type cassette switching, site-specific DNA recombination, meiotic chromosomal recombination, and genome editing, DNA is not a static object but can act dynamically in a programmed manner. Although highly efficient genome editing methods can be used to construct DNA plasmids[27,28] and synthetic genetic circuits[29–32] (both involving DNA sequence changes), no standard framework is proposed to simulate changes in annotated dsDNA sequences. To demonstrate QUEEN can also program, simulate, and share such dynamic DNA circuits, we demonstrated the construction of

a genetic six-input, one-output Boolean Logic Look Up Table (LUT) demonstrated in BLADE[33] using QUEEN (Fig. 5a). In this circuit, upon input of site-specific DNA recombinases, the circuit DNA sequence alters by multistep deletions and/or inversions of segments sandwiched by corresponding recombinase target sequences. The input patterns of four recombinases (Vica, B3, PhiC31, and Bxb1) configures one of the 16 Boolean logic gates, and the remaining two recombinases (Cre and FLP) serve as two input signals to the logic gate (Fig. 5b). The output is given as GFP expression. The intertwined segment recombinations yield one or none of the four GFP genes to be expressed, depending on the signal input pattern.

We implemented DNA segment deletion by "cut" and "join," and inversion by "cut," "flip," and "join." After obtaining a gbk file describing the initial state of the DNA circuit, we loaded it to a QUEEN script and simulated its behaviors for all of the 64 possible signal input patterns (Fig. 5c and Supplementary Table 3). All the input patterns conferred the expected DNA sequence outcomes. We also output the resulting dsDNA objects to gbk files and

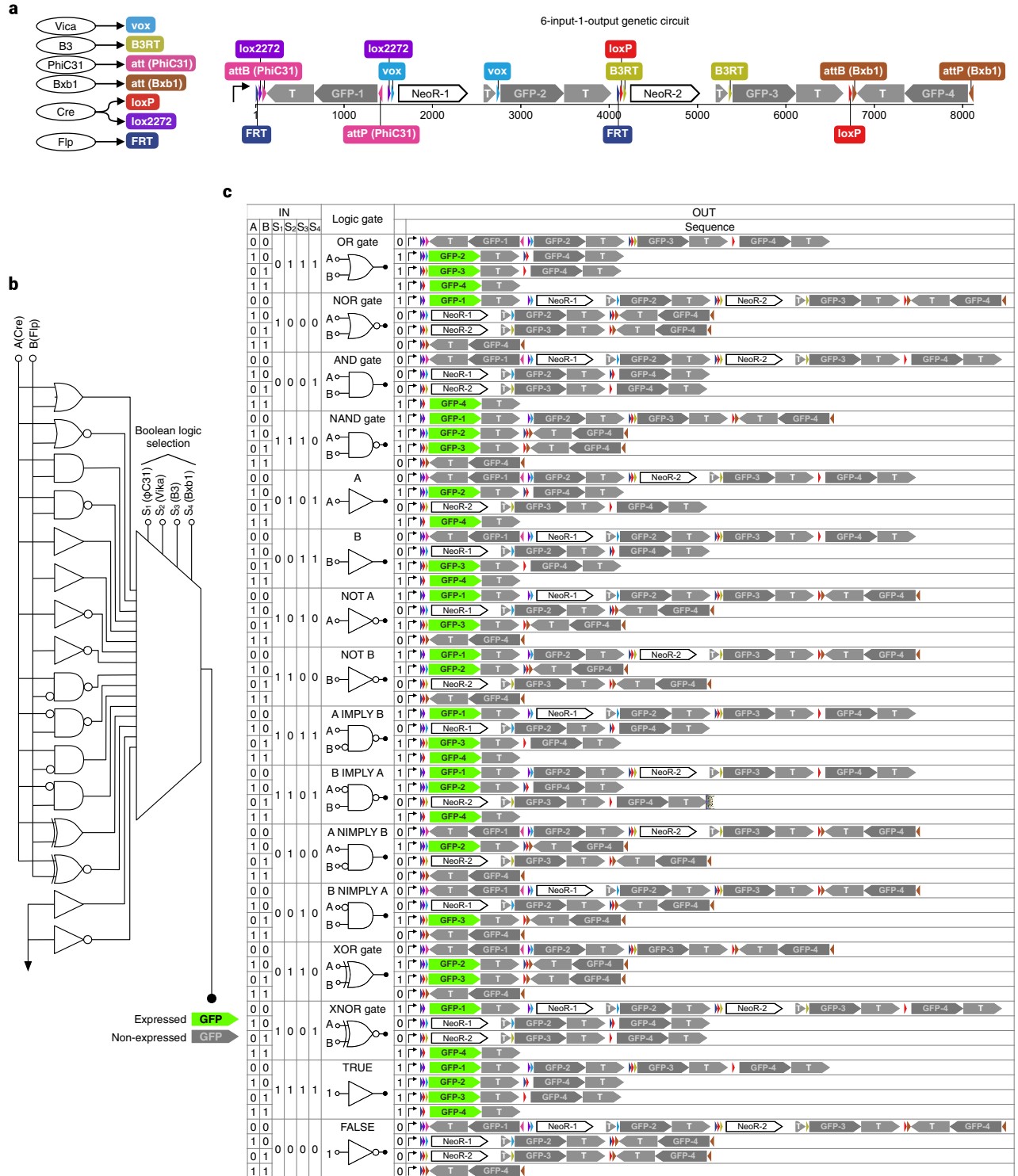

**Fig. 5 Simulation of a Boolean logic LUT. a** The six-input, one-output Boolean logic LUT of the BLADE system. Left, site-specific DNA recombinases (round nodes) and their corresponding target DNA sites (round rectangles). Right, the genetic Boolean logic LUT circuit. **b** The wiring diagram of the genetic circuit. **c** Annotated DNA sequence outcomes of each signal input pattern simulated by QUEEN. GFP expresses if one of the GFP-encoding genes is placed in the same direction as the upstream promoter without being prevented by a transcription terminator.

demonstrated that their quine codes and process histories could be derived from the output files (Supplementary Fig. 4).

## Discussion

There have been several software packages developed to design molecular cloning procedures and generate annotated plasmid files, such as ApE[34], Benchling (https://www.benchling.com/), Geneious[35], j5[36], Pydna[37], Raven[38], and SnapGene (https://www.snapgene.com/). While most of them are GUI-based software tools for local client computers, Benchling is a cloud-based software tool and Pydna is a Python programming package that enables the description of DNA construction processes. Notably, the operational functions of these tools are all implemented for

specific cloning methods, such as Gibson Assembly, Golden Gate Assembly, and traditional RE digestion and ligation cloning. These specified functions seem intuitive and convenient for users as long as they design DNA materials with the prepared set of methods, but lack the elasticity to incorporate new DNA building methods. In contrast, we hypothesized that DNA construction processes can be generalized by the combination of four basic operations, "cut," "flip," "join," and "modify ends." We demonstrated that this system could easily describe the equivalent operations of the previous tools in various examples. Although a range of currently available genomic resources are genetically modified by gene deletion and transgene insertion through homologous DNA repair, Cre-loxP, and genome editing, no software tool to describe all these processes has been developed. The simulation of the BLADE circuit showed that QUEEN is also capable of flexibly describing a process of dynamic DNA sequence alterations. In combination with the existing simulation platforms for transcription unit (TU)-based genetic circuits[39–41], QUEEN could also accelerate the construction of genetic circuits and cells having those.

Two major contrivances were implemented in QUEEN to permeate its use in the life science community: (1) the use of gbk file format and (2) the self-reproducibility of process codes. We first determined that the gbk file format is the foremost format that QUEEN needed to adopt for the life science community that widely uses gbk files to share DNA maps. Furthermore, while SBOL has been proposed to enable organized annotation of structural and functional aspects of DNA sequences in contrast to the gbk file format, we considered it important to demonstrate the concept of QUEEN in the more unorganized semantic system of gbk to showcase its versatility in file format. Recording of self-reproducible operational history in the output gbk file and their inheritance by descendant gbk files are also unique features of QUEEN that have great potential to change the ways of building DNA. The ability to generate a quine code from a QUEEN-generated gbk file not only certifies its reproducibility but also enables the accompaniment of protocols and those of parental DNA materials with the generating gbk file. This is the key feature of QUEEN that enables the community to share DNA materials and protocols together without asking for the additional cost of consciously managing these two separately. A protocol retrieved from a QUEEN-generated gbk file can be edited to generate a new DNA construct. This process can also be recorded and passed to a newly producing gbk file. Benchling and SnapGene are also capable of recording DNA construction processes. However, they are unable to easily share the construction protocols outside of the software environments nor track the deep inheritance of materials and protocols from one DNA construct to another.

If the life science community starts generating and sharing gbk files using QUEEN or QUEEN-compliant software tools, there will be new ways of evaluating resources and developers' contributions. The impacts of materials and protocols can be evaluated not only by a "trending" information of how many times they are requested by other researchers, like shown in Addgene, but also by how they are inherited in successive products even over multiple generations. However, the current implementation of QUEEN for the recovery of process histories from their gbk files has two potential issues. First, as the community starts utilizing QUEEN-generated gbk files, the sizes of new QUEEN-generated gbk files will keep increasing. Second, the current QUEEN framework highly depends on the community's goodwill and cannot certify original developers of DNA materials or that the utilization of previously created resources is properly recorded in QUEEN-generated gbk files. For cell strains, CellRepo has proposed for the deposition of engineered strains with specific DNA barcode identifiers where CellRepo serves as a certification authority[42]. While the current issues of QUEEN could be resolved by a similar cloud-based certification authority system that authorizes, stores, and traces DNA material and protocol resources together with QUEEN-compliant client software tools, such a framework would not be most effective until the sharing of both DNA material and protocol resources is widely communized with the current QUEEN framework.

Once QUEEN is widely adopted in the life science community, it would also accelerate the development of an efficient DNA design system. There have been algorithms proposed to compute efficient DNA assembly steps for target DNA products from a given synthetic DNA library resource[36,38,43]. The wealth of practical DNA construction process knowledge made available by QUEEN will greatly contribute to such an automated design of DNA construction processes, where a building process of user-requested DNA can be autonomously designed with the most optimal recycling of available DNA materials and reagents in a user's environment, and with practically the best DNA cloning strategy chosen based on the knowledge of how many times the community has succeeded in similar methods. This idea could also be implemented to gene synthesis and assembly automation systems[44].

While challenging, it is also important to establish similar systems to share materials and reproducible protocols in other experimental domains that require more complex descriptions. Laboratory automation of natural science experiments in general requires the full semantic description of reproducible protocols in a robot-executable manner. Ideally, this would resolve many of the current issues in life science, including the reproducibility crisis[45]. However, significant technical and social contrivances would be needed to realize this goal primarily because the development of robotic systems requires tremendous investments, and laboratory automation communities have not been coherently progressing (different projects develop their own systems which cannot easily be integrated). As represented by Protocol Activity Modeling Language (PAML) (https://github.com/Bioprotocols/paml), several process ontologies have been proposed to describe life science experimental processes. However, we believe good practices like the ones implemented in QUEEN are important to motivate the current life science community to adopt a new system without feeling a substantial cost. A potential direction for reproducible process sharing and laboratory automation of various experimental domains could be the development of a standardized "multiscale" process semantics, where any experimental process of any resolution can be programmed, from the resolution of current material and methods descriptions with researchers' best efforts to that of robotic executions with their APIs. It may be possible to encourage the community to first use such a semantics for daily experiments with the support of GUI-assisted editors, whose description can be shared, reused, and elaborated later for laboratory automation with robotics. If widespread, this could accelerate the momentum of laboratory automation towards a similar vision that we propose here with QUEEN. Accordingly, we report QUEEN as a framework to accelerate describing and sharing of DNA material and protocol resources, and as a starting point to ponder a similar system for entire life sciences domains with laboratory automation.

## Methods

**Implementation of QUEEN**. QUEEN was implemented as a Python 3.7 module. It requires BioPython[46] mainly for GenBank file parsing. The visualization of annotated DNA sequence maps (visualizemap) and operational process flow charts (visualizeflow) from QUEEN objects are dependent on Python Matplotlib[47] and Graphviz[48] modules, respectively. The detailed usage of each function implemented in the QUEEN module is described in the GitHub repository (https://github.com/yachielab/QUEEN/blob/master/README.md). Example QUEEN scripts are provided as Jupyter Notebook files at https://github.com/yachielab/QUEEN/tree/master/demo/tutorial.

**Reporting summary**. Further information on research design is available in the Nature Research Reporting Summary linked to this article.

## Data availability

Of the gbk files used as input files for the simulation of base editor plasmid constructions, pLV-eGFP, pCMV-ABE7.10, pcDNA3.1_pCMV-nCas-PmCDA1-ugi pH1-gRNA(HPRT), pCMV-BE4max, and pCMV-ABEmax were obtained from Addgene (Plasmid IDs: 36083, 102919, 79620, 112903, and 112905, respectively). The sequence file for pLV-SIN-CMV-Puro was obtained from Takara Bio, Inc. (Japan, https://catalog.takara-bio.co.jp/DNA_seq/pLVSIN-CMV_pur.zip). The gbk file for pRS112 and pUC-optimized-PmCDA1-ugi encoding the codon-optimized PmCDA1-UGI was created using Benchling. Some detail sequence feature annotations of input files were added manually before using them for the demonstration (the modified files are available at https://github.com/yachielab/QUEEN/tree/master/demo/sakata_et_al_2020). The gbk file used for the simulation of the Boolean logic LUT circuit was downloaded from Addgene (Plasmid ID 87554), to which sequence feature annotations for the site-specific recombination sites were added manually before the demonstration (the modified file is available at https://github.com/yachielab/QUEEN/tree/master/demo/Weinberg_et_aL_2017).

## Code availability

QUEEN is an open-source software package distributed with MIT License. The entire package, installation, and user's manual are available at the GitHub repository (https://github.com/yachielab/QUEEN/). All of the source codes for QUEEN are placed in https://github.com/yachielab/QUEEN/tree/master/QUEEN. The QUEEN scripts used to construct the base editor plasmids and the simulation of the Boolean logic LUT are provided as Jupyter Notebook files at https://github.com/yachielab/QUEEN/tree/master/demo/sakata_et_al_2020 and https://github.com/yachielab/QUEEN/tree/master/demo/Weinberg_et_aL_2017, respectively. All of the Jupyter Notebook files for the demonstrations in this study are available in GitHub and made executable in Google Colaboratory (Supplementary Table 1).

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

## Acknowledgments
We thank the members of the Yachie laboratory at the University of British Columbia and the University of Tokyo for useful comments and discussions, especially Soh Ishiguro for testing the code and Samuel King for proofreading and providing feedback on the manuscript. This study was performed under the Canada Research Chair program supported by the Canadian Institutes for Health Research (CIHR). H.M. was supported by the JSPS DC2 Fellowship and TTCK Fellowship (from the Yamagata prefectural government and Tsuruoka city).

## Author contributions
H.M. and N.Y. conceived the study and designed QUEEN. H.M. implemented QUEEN. H.M. and N.Y. wrote the manuscript together.

## Competing interests
The authors declare no competing interests.
