## [Peer Review File · Nature Communications]

Reviewers' Comments:

Reviewer #1:

Remarks to the Author:

The authors introduce QUEEN, an extension to the GenBank file format (gbk files for short) with an associated toolset, which enables the specification of DNA constructs (e.g. single-stranded polymers, double-stranded polymers or circular plasmids) in a way that is (a) self-defining, (b) more reproducible and (c) compatible with tools that already handle gbk format.

The authors argue that the current practice of DNA specification (which takes place before synthesis) is haphazard, error-prone, mostly done in a manual manner via graphical user interfaces and seldom leads itself to resource reuse or credit allocation. I agree with the authors on these points and on the timeliness of their approach.

The authors present what, in essence, is a small domain-specific programming language (DSL) that allows specifying the series of editing operations one may perform on a DNA polymer in order to fully define it. Their DSL allows for re-use of previously designed DNA fragments and lend itself to capturing the editing history of the specified molecule, including keeping track of the source of the various fragments.

The main innovation presented is the extension to the GenBank file format to also capture, as part of its "FEATURES" section, a set of domain-specific language directives (QUEEN functions) that describe the process that created the DNA sequence captured in the GenBank file. That is, the authors effectively turn sequences described in GenBank files into self-defining ones. This is a very clever insight.

The domain-specific language introduced enables also the tracking of DNA fragments reuse and builds a history of how -in turn- those fragments themselves were defined. One way to look at this is to visualise the quine code as a "blockchain" of previous DNA operations, except that, in this case, there is no proof-of-work or proof-of-stake involved, which makes it fast and computationally inexpensive.

The authors also provide various tools to graphically interpret the information stored in the QUEENed GenBank file and these tools are welcomed.

I find the idea of using a self-defining GenBank file compelling and convincing. Having downloaded the code, tested it and also gone through the Jupyter notes provided, I believe that the authors have done a great job in making this work reproducible and reusable.

There are a few weaknesses in the paper that the authors could address and that I hope might make the paper a bit more life-sciences friendly. Currently, it reads more like a computer science-oriented one and I think if the objective is wide-range adoption, lowering the barrier by improving the text might help. Thus, here are some points the authors may want to consider:

- The title is just too generic, it does not reflect well what the paper and its tools actually do. I would encourage the authors to find a more specific and targeted title.
- The paper is verbose and quite dense in some parts and I suspect that readers without a strong computing science background might get lost with all the definitions and discussions about quines, etc. I am uncertain all of that is necessary. In fact, I believe it is enough to simply say that the GenBank file is extended to include the code that defines the editing process used on the DNA sequence stored in the gbk file; then simply say that by having the sequence and the process together you can check that one derives from the other.
- Clearer pictures. Figure 1 is ok-ish but delves too much on what may happen if people adopt QUEEN extensively rather than actually dealing more clearly with explaining the concept of a self-defining file. Figure 3 is just too complex, maybe it can be replaced with a link to a "live" version of the figure on the web where one could interact, pan, zoom, etc to understand it.

- The jupyter notes, although quite exhaustive (thank you, I appreciated that!), could do with a bit simpler language explanation of what is going on to guide a perhaps not too familiar with python (let alone QUEEN) life-science reader.
- There are a number of discussions of what may or may not happen in the future if people were to adopt QUEEN that could be reduced and better articulated.
- There are a number of claims that are either wrong, just too general or not substantiated by the work presented. For example, between lines 58 and 68 and then again between lines 257 and 270 the authors should note that previous work does indeed exist along the lines discussed. For example [1] presents a specialised version control system for strain engineering while [2] already introduced a domain-specific language for combinatorial DNA library specification that included sequences re-use, construction planning (and NP-hardness related proof) and version control via its IDE of the DNA library specification. True, [2] did not include "self-definition" like QUEEN does but was also generic in the sense that it was not protocol (e.g. Golden Gate, Gibson, etc) specific.
- The authors should discuss further the implication of having rapidly growing GenBank files. For example, I downloaded from New England Biolabs a raw (i.e. without additional QUEEN directives) pUC19.gbk sequence and tested the code by generating a new gbk file for pUC19 via the QUEEN class but without modifying the sequence:

```
pUC19 = QUEEN(record="./pUC19.gbk")
iteration1 = QUEEN(record="./pUC19.gbk")
iteration2 = QUEEN(record="./it1.gbk")
iteration3 = QUEEN(record="./it2.gbk")
iteration4 = QUEEN(record="./it3.gbk")
```

and although the sequence itself did not change across iterations as there were no additional QUEEN directives applied, the size of the file grew by 510 bytes after 4 iterations. Cannot this be optimised?

- The argument that QUEEN will eliminate mis-crediting DNA resources/protocols creators is not, strictly speaking, correct. As I mentioned earlier keeping track of the DNA editing process in the GenBank file is akin to creating a "blockchain" of previous DNA operations, except that because the "ledger" is not distributed but rather centralised on the QUEEN file itself and because there is no proof-of-work or proof-of-stake involved a malicious actor could still cheat.
- Finally, I wonder why the authors opted for the more stringent GNU General Public Licence V3.0 rather than the more permissive e.g MIT one. If the objective is wide and rapid dissemination, shouldn't the license be more permissive?

References:

[1] Versioning Biological Cells for Trustworthy Cell Engineering.
 bioRxiv 2021.04.23.441106; doi: <https://doi.org/10.1101/2021.04.23.441106>

[2] ACS Synth. Biol. 2014, 3, 8, 529–542. <https://doi.org/10.1021/sb400161v>
 Code: <http://dnald.ico2s.org/>

Reviewer #2:

Remarks to the Author:

In this work, Mori & Yachie have developed QUEEN, which is a framework to share material and resources for DNA construction. While the concept is interesting and QUEEN may be useful, the reviewer was wondering whether it would have a real utility in the field that the journal readers expect. The concept of standardization in synthetic biology is not novel, but it has been proposed by multiple researchers more than a decade ago (e.g., Nature Biotechnology volume 26, pages 787–793 (2008), which is missing in the reference). Additionally, we have already observed computation-based genetic circuit building multiple times, culminating in the seminal paper (Science, 2016 Vol 352, Issue 6281; DOI: 10.1126/science.aac7341, which is also missing in the reference). While these advances may be indirectly related to the authors' work, their limited adoption by general synthetic biology community researchers might indicate or imply the limited utility of the current work. In contrast to those previous reports that either proposed the new concept (i.e., synthetic biology framework such as standardization of interchangeable parts) for the first time, or experimentally demonstrated algorithm-guided complex circuit building, the current work seems limited in technological advances or the scope. Thus, publication in this high-impact journal would be premature, although the approach and the framework look sound.

Reviewer #3:

Remarks to the Author:

This paper presents, QUEEN, which is effectively a python procedure for manipulating DNA sequence assemblies. Replacing natural language descriptions of assembly plans with a programmatic and reproducible solution is an admirable goal. The authors discuss that this would enable a more optimized approach to assembly planning, which is correct though not covered by this paper.

While the work is promising and may prove to eventually be useful, it is currently flawed due to its use of the GenBank file format. Yes, GenBank is flexible, as the authors point out, but this is its fundamental flaw. The annotations they put in the GenBank files are not standardized, so they will not be meaningful to any tool but their own. This makes the resulting files not substantially improved over natural language. GenBank is also a flat file format, so using this format loses the hierarchical nature of a design assembled from parts.

Rather than using GenBank and BioPython, the authors should instead use the Synthetic Biology Open Language (SBOL) and pySBOL. SBOL is a community developed standard that is capable of capturing not only sequence annotations, but also preserve the hierarchical nature of an assembled genetic circuit. It also also capable of describing the entire assembly process in one document rather than a collection of GenBank files and a Python script. Indeed, once the SBOL is constructed, the script is not absolutely required to reconstruct the assembly process. SBOL also uses the Provenance Ontology (Prov-O) to represent the metadata for the provenance of the assembly.

In order to represent the actual protocols, the authors are encouraged to check out the community developed Process Activity Modeling Language (PAML).

Response to reviewers' comments

For “A framework to efficiently describe and share reproducible protocols in the DNA construction world” (NCOMMS-21-44872A)

Revision Summary:

We sincerely appreciate the constructive peer-review process to improve our manuscript and new framework QUEEN to share DNA materials and construction protocols efficiently through the life science community. We thoroughly addressed the comments raised by the reviewers and updated the manuscript.

In summary, Reviewer 1 agreed with the impact of the work but requested to substantially revise the manuscript so that it communicates well with readerships in the biology domain, by avoiding redundant, verbose and computer science (CS)-oriented explanations, and some inaccurate interpretations of the work.

After reading the comments by Reviewers 2 and 3, we realized that the way we originally structured the manuscript was misleading, and we were not properly delivering our main idea and achievements with the context of what has been achieved in the field of synthetic biology. We revised the manuscript to explain the current work in contrast to the previous efforts as follows.

In brief, the current standardizations of DNA materials have focused on two aspects (lines 41–55): (1) the modularization of DNA parts which can be seen in MoClo (PMID: 21364738) and BioBrick (PMID: 18410688), and (2) the description of functional DNA parts that has been achieved by the Synthetic Biology Open Language (SBOL; PMID:24911500).

In this work, we propose a third type of semantic for (3) the standardization of “DNA construction processes” (lines 56–65). We carefully explained (a) the design of the simple semantics, (b) its implementation to best encourage and benefit the life science community, and (c) potential impacts of the new framework from different aspects.

In brief,

- (a) We propose that any DNA construction processes can be described with four simple operational functions, two search functions and another set of functions that directly edit DNA sequences and their annotations (lines 101–137).
- (b) We also implemented it as a Python programming package that generates output DNA into a GenBank (gbk) format file, which also encodes the Python codes that generated itself (lines 77–98). This “quine” function of QUEEN allows end-users to start sharing DNA material resources together with their construction processes by sharing them just like sharing regular gbk files. The ability to generate a quine code from a QUEEN-generated gbk file not only certifies its reproducibility but also enables the accompaniment of protocols and those of parental DNA materials within the generating gbk file. This is the key feature of QUEEN that enables the community to share DNA materials and protocols together without asking for the additional cost of consciously managing these two separately (lines 290–294).
- (c) We also want to primarily discuss three potential impacts of the new framework. First, QUEEN enables a new gbk file to inherit all parental DNA material information and their full construction processes. This enables a new way of evaluating impacts of DNA materials and their protocols in the life science community (lines 151–157). Second, the description of materials and methods can be replaced by this process semantic language, or the completeness

of materials and methods written in natural language can be confirmed along with the process semantics (lines 157–161). (QUEEN enables users to create a gbk file that can produce a Materials and Method description.) Third, by accelerating the life science community to easily accumulate DNA construction process resources, we can envision establishing an AI-based system to design an efficient DNA construction strategy for a target DNA product, given the available DNA material and reagent information in a given environment, and laboratory automation of such (lines 315–323).

Reviewer 3 also suggested to use SBOL instead of the GenBank (gbk) format to describe annotated DNA structure. However, we first determined that the gbk file format is the foremost format that QUEEN needed to adopt for the life science community that widely uses gbk files to share DNA maps (lines 282–285). Furthermore, while SBOL has been proposed to enable organized annotation of structural and functional aspects of DNA sequences in contrast to the gbk file format, we considered that it is important to demonstrate the concept of QUEEN in the more unorganized semantic system of gbk to showcase its versatility in file format (lines 285–288).

We clarified these points and novelty of the work in the revised manuscript. We would sincerely appreciate it if the reviewer could kindly examine the work again.

Reviewers' comments:

Reviewer #1

Comment 1.1:

(Remarks to the Author):

The authors introduce QUEEN, an extension to the GenBank file format (gbk files for short) with an associated toolset, which enables the specification of DNA constructs (e.g. single-stranded polymers, double-stranded polymers or circular plasmids) in a way that is (a) self-defining, (b) more reproducible and (c) compatible with tools that already handle gbk format.

The authors argue that the current practice of DNA specification (which takes place before synthesis) is haphazard, error-prone, mostly done in a manual manner via graphical user interfaces and seldom leads itself to resource reuse or credit allocation. I agree with the authors on these points and on the timeliness of their approach.

The authors present what, in essence, is a small domain-specific programming language (DSL) that allows specifying the series of editing operations one may perform on a DNA polymer in order to fully define it. Their DSL allows for re-use of previously designed DNA fragments and lend itself to capturing the editing history of the specified molecule, including keeping track of the source of the various fragments.

The main innovation presented is the extension to the GenBank file format to also capture, as part of its “FEATURES” section, a set of domain-specific language directives (QUEEN functions) that describe the process that created the DNA sequence captured in the GenBank file. That is, the authors effectively turn sequences described in GenBank files into self-defining ones. This is a very clever insight.

The domain-specific language introduced enables also the tracking of DNA fragments reuse and builds a history of how--in turn--those fragments themselves were defined. One way to look at this is to visualise the quine code as a “blockchain” of previous DNA operations, except that, in this case, there is no proof-of-work or proof-of-stake involved, which makes it fast and computationally inexpensive.

The authors also provide various tools to graphically interpret the information stored in the QUEENed GenBank file and these tools are welcomed.

I find the idea of using a self-defining GenBank file compelling and convincing. Having downloaded the code, tested it and also gone through the Jupyter notes provided, I believe that the authors have done a great job in making this work reproducible and reusable.

There are a few weaknesses in the paper that the authors could address and that I hope might make the paper a bit more life-sciences friendly. Currently, it reads more like a computer science-oriented one and I think if the objective is wide-range adoption, lowering the barrier by improving the text might help. Thus, here are some points the authors may want to consider:

Response 1.1:

Thank you very much for all the encouraging and constructive comments. We have revised the manuscript according to the reviewer’s suggestions. Please see our point-by-point responses below.

Comment 1.2:

The title is just too generic, it does not reflect well what the paper and its tools actually do. I would encourage the authors to find a more specific and targeted title.

Response 1.2:

Yes, it might sound too generic. In the revised version, we slightly modified the title to “A framework to efficiently describe and share reproducible protocols in the DNA construction world,” which is more specific to the framework we established. (The previous title was “A universal framework to efficiently share material and process resources in the DNA construction world.”) We also realized that we need to highlight that we developed a new way of describing and storing protocol resources, but this has nothing to do with how to annotate DNA resources, so the title was changed accordingly.

We believe that the key achievements of the new framework are not only that it enables a GenBank (gbk) format file to self-clone its construction process (we sincerely appreciate that the reviewer resonated with this concept), but also that it enables the efficient design and sharing of DNA materials. We believe the potential impacts of QUEEN that we have discussed in the manuscript are high and that the same idea should, ideally, be implemented into other experimental domains. Therefore, we would like to propose this new title; although, another potential title could be “A framework to efficiently describe and share reproducible DNA construction protocols”, but this might sound a bit tedious. We would love to get further feedback.

Comment 1.3:

The paper is verbose and quite dense in some parts and I suspect that readers without a strong computing science background might get lost with all the definitions and discussions about quines, etc. I am uncertain all of that is necessary. In fact, I believe it is enough to simply say that the GenBank file is extended to include the code that defines the editing process used on the DNA sequence stored in the gbk file; then

simply say that by having the sequence and the process together you can check that one derives from the other.

Response 1.3:

Thank you very much. We agree that the initial manuscript had redundant statements with too much detail or CS-oriented descriptions on how to use QUEEN. This was mainly because we wanted to ensure that the new concept could be precisely delivered to readers. However, we now agree with the reviewer's comment that the initial manuscript was simply too verbose, did not flow well, and also a bit confusing to the readership with a limited computational background. At the same time, we realized that QUEEN highly relies on the new syntax we defined for QUEEN (including the four simple operational functions: cut, modify, flip, and join). Therefore, we maintained the high-level concept of some major functions of QUEEN in the revised manuscript (lines 101–163). We omitted the command names and other miscellaneous functions from the initial manuscript, and instead describe them in the user manual we provide (this was also a suggestion we obtained from our colleagues when we asked them for feedback). We hope the revised version now reads well without detracting from the key ideas implemented in the framework.

Comment 1.4:

Clearer pictures. Figure 1 is ok-ish but delves too much on what may happen if people adopt QUEEN extensively rather than actually dealing more clearly with explaining the concept of a self-defining file.

Response 1.4:

We appreciate this comment. We agree that Figure 1 was too busy. We thought the potential impacts of the new framework when people adopt QUEEN is as important as the new concept of a self-defining file. We still want to keep these figure panels, but we also realized that the other panel explaining QUEEN's functions in the original figure was too large and diluted the main concept. Therefore, in this revised version, we instead changed the panel explaining the functions to an independent main figure (Figure 2) and simplified Figure 1.

Comment 1.5:

Figure 3 is just too complex, maybe it can be replaced with a link to a “live” version of the figure on the web where one could interact, pan, zoom, etc to understand it.

Response 1.5:

We also appreciate this comment. The original Figure 3 is now Figure 4 in the revised manuscript. According to the reviewer's suggestion, we moved panel “b” to Benchling, which not only allows the readership to interact with the figure but also demonstrates that QUEEN-generated files can be operated using a common plasmid editing software tool (<https://benchling.com/s/seq-70uRXXVhQiANrwkhIReK?m=sIm-9cOQEy1QQppXpyDIxyzI>; the same URL is also listed in Supplementary Table 2). Panel “a” demonstrates a concrete example that QUEEN can trace multi-step DNA fragment resource inheritances. The current panel “b” (previously Figure 3c) demonstrates the autonomous recovery of a sequence feature annotation for a sequence that was once fragmented during the DNA construction process but recovered in the final plasmid product. We appreciate if the reviewer could agree to keep these key demonstrations.

Comment 1.6:

The jupyter notes, although quite exhaustive (thank you, I appreciated that!), could do with a bit simpler language explanation of what is going on to guide a perhaps not too familiar with python (let alone QUEEN) life-science reader.

Response 1.6:

Thank you very much! In addition to the tutorials we prepared (tutorial_ex01-23.ipynb and tutorial_ex24-28.ipynb), we have now provided a simple narrative explanation in comments on each operational line of another two representative demonstrations (pCMV_Target_AID_construction.ipynb and Boolean_logic_LUT.ipynb). The URLs to the Jupyter Notebooks and Google Colaboratory repositories are listed in Supplementary Table 1.

Comment 1.7:

There are a number of discussions of what may or may not happen in the future if people were to adopt QUEEN that could be reduced and better articulated.

Response 1.7:

Yes, these discussions were also redundant. In the revised manuscript, we limited our discussion on the potential community impacts in the last two paragraphs of Discussion (lines 315–344).

Comment 1.8:

There are a number of claims that are either wrong, just too general or not substantiated by the work presented. For example, between lines 58 and 68 and then again between lines 257 and 270 the authors should note that previous work does indeed exist along the lines discussed. For example [1] presents a specialised version control system for strain engineering while [2] already introduced a domain-specific language for combinatorial DNA library specification that included sequences re-use, construction planning (and NP-hardness related proof) and version control via its IDE of the DNA library specification. True, [2] did not include “self-definition” like QUEEN does but was also generic in the sense that it was not protocol (e.g. Golden Gate, Gibson, etc) specific.

References:

- [1] Versioning Biological Cells for Trustworthy Cell Engineering. bioRxiv 2021.04.23.441106; doi: <https://doi.org/10.1101/2021.04.23.441106>
- [2] ACS Synth. Biol. 2014, 3, 8, 529–542. <https://doi.org/10.1021/sb400161v>
Code: <http://dnald.ico2s.org/>

Response 1.8:

We sincerely appreciate this comment to correct and improve our manuscript. We realized that both suggested papers were very valuable to discuss QUEEN. We clarified the following points in the manuscript.

The first suggested paper (now published in *Nature Communications*; PMID: 35140226) describes a GitHub-like system called CellRepo to deposit information of engineered cell strains with specific DNA identifiers, where CellRepo serves as a certification authority. The idea described in this system greatly complements the idea of QUEEN and provides insights to strategize sharing protocols and materials of DNA resources. For example, as the reviewer suggested, QUEEN realizes a block chain-like system for the sharing of gbk files, where previous construction histories of DNA materials used

in the current DNA construction process are all encoded into the producing gbk file. We focused on this standalone file-based system that does not require a third certification authority so the QUEEN-generated (and therefore QUEEN-compatible) gbk files can be easily disseminated to the community without particularly recognizing its usage and need in the beginning (lines 290–294). (We are currently discussing with Addgene their plan to put a slight modification in their system such that QUEEN-generated gbk files will be available with clear notations that they are QUEEN-generated/compatible.) However, in the long term, this may end up ballooning the data sizes of gbk files by the repetitive inheritance of DNA resources from previous DNA products. A cloud-based certification authority system storing such process history data and communicating with QUEEN would greatly mitigate this issue and maintain compact gbk file sizes (lines 305–306). Furthermore, the current QUEEN framework highly depends on the community's goodwill and cannot certify the original developer of DNA materials or that the utilization of previously created resources are properly recorded in QUEEN-generated gbk files (lines 307–309). We discussed that a system like CellRepo could mitigate this issue too, but we also mentioned that such a framework might not be most effective until the sharing of both DNA material and protocol resources are widely communized with the current QUEEN framework (lines 309–314).

The second suggested paper (PMID: 24730371) describes a method to compute the efficient construction of a target DNA sequence from available DNA resources with the shortest number of steps. One of the major motivations in the development of QUEEN was the need for a public process semantic system to describe DNA construction processes, and the community-based accumulation of such process resources to better design new DNA parts, not only from the end inputs and outputs, but also with the wealth of knowledge of processes that can be achieved by QUEEN. This would accelerate the development of an AI-based laboratory automation system that designs and constructs new DNA sequences efficiently (lines 315–323). We referred to the paper and briefly discussed these ideas.

Comment 1.9:

The authors should discuss further the implication of having rapidly growing GenBank files. For example, I downloaded from New England Biolabs a raw (i.e. without additional QUEEN directives) pUC19.gbk sequence and tested the code by generating a new gbk file for pUC19 via the QUEEN class but without modifying the sequence:

```
pUC19 = QUEEN(record="./pUC19.gbk")
iteration1 = QUEEN(record="./pUC19.gbk")
iteration2 = QUEEN(record="./it1.gbk")
iteration3 = QUEEN(record="./it2.gbk")
iteration4 = QUEEN(record="./it3.gbk")
```

and although the sequence itself did not change across iterations as there were no additional QUEEN directives applied, the size of the file grew by 510 bytes after 4 iterations. Cannot this be optimised?

Response 1.9:

Thank you very much for testing QUEEN! As responded in Response 1.8, we now discuss the potential rapid growth of GenBank file sizes with the need for adopting a cloud-based certification authority system (lines 305–314).

We were actually unaware of the slight file size increase with the cloning of gbk files by QUEEN. We replicated the same phenomenon, but soon realized that this was because the cloned gbk files encode the previous histories of cloning iterations and we hope the reviewer can agree that this is OK. We can modify the code not to leave the operation process history when a user just clones the original file, but we thought it is probably better to keep it as it is.

Comment 1.10:

The argument that QUEEN will eliminate mis-crediting DNA resources/protocols creators is not, strictly speaking, correct. As I mentioned earlier keeping track of the DNA editing process in the GenBank file is akin to creating a “blockchain” of previous DNA operations, except that because the “ledger” is not distributed but rather centralised on the QUEEN file itself and because there is no proof-of-work or proof-of-stake involved a malicious actor could still cheat.

Response 1.10:

This is very correct, and we should have been more careful in writing the manuscript. Basically, QUEEN highly depends on the goodwill of the life science community, like citing papers when writing a paper, where people can cheat. This issue can be alleviated when the idea of a cloud-based certification authority is implemented to certify QUEEN-generated products (lines 307–314). We carefully revised our statements in this regard.

Comment 1.11:

Finally, I wonder why the authors opted for the more stringent GNU General Public Licence V3.0 rather than the more permissive e.g MIT one. If the objective is wide and rapid dissemination, shouldn't the license be more permissive?

Response 1.11:

Thank you very much. We originally thought that the software packages using QUEEN should also be open-source. However, as we started considering the development of GUI-based software tools with QUEEN (for example, we have started discussing with Benchling the possibility of implementing QUEEN into their platform), we have since changed the license to the MIT one.

Reviewer #2

Comment 2.1:

(Remarks to the Author):

In this work, Mori & Yachie have developed QUEEN, which is a framework to share material and resources for DNA construction. While the concept is interesting and QUEEN may be useful, the reviewer was wondering whether it would have a real utility in the field that the journal readers expect. The concept of standardization in synthetic biology is not novel, but it has been proposed by multiple researchers more than a decade ago (e.g., Nature Biotechnology volume 26, pages 787–793 (2008), which is missing in the reference).

Response 2.1:

Thank you very much for the valuable review comments. Since it has been a while since we have received the reviewers' comments, we provided a Revision Summary in the beginning of this document. We hope that this will help the reviewer remember the content of the paper and digest the responses below.

After reading the comments by Reviewers 2 and 3, we realized that the way we originally structured the manuscript was misleading, and we were not properly delivering our main idea and achievements with the context of what has been achieved in the field of synthetic biology. We revised the manuscript to explain the current work in contrast to the previous efforts as follows.

As described in Revision Summary in the top page of this document, the current standardizations of DNA materials have focused on two aspects (lines 41–55): (1) the modularization of DNA parts which can be seen in MoClo (PMID: 21364738) and BioBrick (PMID: 18410688) and (2) the description of functional DNA parts that has been achieved by the Synthetic Biology Open Language (SBOL; PMID: 24911500).

In this work, we propose a third type of semantic for (3) the standardization of “DNA construction processes” (lines 56–65). We carefully explained (a) the design of the simple semantics, (b) its implementation to best encourage and benefit the life science community, and (c) potential impacts of the new framework from different aspects.

We clarified these points and novelty of the work in the revised manuscript. We sincerely appreciate if the reviewer could kindly examine the work once again.

Comment 2.2:

Additionally, we have already observed computation-based genetic circuit building multiple times, culminating in the seminal paper (Science, 2016 Vol 352, Issue 6281; DOI: 10.1126/science.aac7341, which is also missing in the reference).

Response 2.2:

As suggested by the reviewer, there have been frameworks developed to design and simulate genetic circuits, most of which are for transcription unit (TU)-based genetic circuits that do not accompany the alteration of DNA sequences. On the other hand, there have recently been attempts to regulate cell systems by dynamically editing DNA sequences, and such attempts encode TU programs using site-specific DNA recombination (PMID: 27463678, PMID: 28346402) and CRISPR genome editing (PMID: 29449507, PMID: 31442423, <https://www.biorxiv.org/content/10.1101/2021.11.05.467388v1>, <https://www.biorxiv.org/content/10.1101/2021.11.05.467434v1>). In Figure 5 (previously Figure 4), we demonstrated that QUEEN has a unique potential to simulate such circuits with DNA alterations. We confirmed that we cited the Cello paper (PMID: 27034378) and clarified these points in the revised manuscript (lines 269–281).

Comment 2.3:

While these advances may be indirectly related to the authors’ work, their limited adoption by general synthetic biology community researchers might indicate or imply the limited utility of the current work. In contrast to those previous reports that either proposed the new concept (i.e., synthetic biology framework such as standardization of interchangeable parts) for the first time, or experimentally demonstrated algorithm-guided complex circuit building, the current work seems limited in technological advances or the scope. Thus, publication in this high-impact journal would be premature, although the approach and the framework look sound.

Response 2.3:

Thank you very much. As described in Revision Summary in the top page of this document, we have clarified the novelty of the work in contrast to the previous efforts in synthetic biology. We hope that the reviewer can now agree with the significance of the work and that this work would be of high interest to the broad life science community.

Reviewer #3**Comment 3.1:**

(Remarks to the Author):

This paper presents, QUEEN, which is effectively a python procedure for manipulating DNA sequence assemblies. Replacing natural language descriptions of assembly plans with a programmatic and reproducible solution is an admirable goal. The authors discuss that this would enable a more optimized approach to assembly planning, which is correct though not covered by this paper.

While the work is promising and may prove to eventually be useful, it is currently flawed due to its use of the GenBank file format. Yes, GenBank is flexible, as the authors point out, but this is its fundamental flaw. The annotations they put in the GenBank files are not standardized, so they will not be meaningful to any tool but their own. This makes the resulting files not substantially improved over natural language. GenBank is also a flat file format, so using this format loses the hierarchical nature of a design assembled from parts.

Rather than using GenBank and BioPython, the authors should instead use the Synthetic Biology Open Language (SBOL) and pySBOL. SBOL is a community developed standard that is capable of capturing not only sequence annotations, but also preserve the hierarchical nature of an assembled genetic circuit. It also also capable of describing the entire assembly process in one document rather than a collection of GenBank files and a Python script. Indeed, once the SBOL is constructed, the script is not absolutely required to reconstruct the assembly process. SBOL also uses the Provenance Ontology (Prov-O) to represent the metadata for the provenance of the assembly.

In order to represent the actual protocols, the authors are encouraged to check out the community developed Process Activity Modeling Language (PAML).

Response 3.1:

Thank you very much for the valuable review comments. Since it has been a while since we have received the reviewers' comments, we provided a Revision Summary in the beginning of this document. We hope that this will help the reviewer remember the content of the paper and digest the responses below.

After reading the comments by Reviewers 2 and 3, we realized that the way we originally structured the manuscript was misleading and we were not properly delivering our main idea and achievements with the context of what has been achieved in the field of synthetic biology. We revised the manuscript to explain the current work in contrast to the previous efforts as follows.

As described in Revision Summary in the top page of this document, the current standardizations of DNA materials have focused on two aspects (lines 41–55): (1) the modularization of DNA parts which can be seen in MoClo (PMID: 21364738) and BioBrick (PMID: 18410688) and (2) the description of

functional DNA parts that has been achieved by the Synthetic Biology Open Language (SBOL; PMID: 24911500).

In this work, we propose a third type of semantic for (3) the standardization of “DNA construction processes” (lines 56–65). We carefully explained (a) the design of the simple semantics, (b) its implementation to best encourage and benefit the life science community, and (c) potential impacts of the new framework from different aspects.

We hope that these collectively answer the first question on why we developed a new Python package for the standardized description of DNA construction processes.

There were two reasons why we adopted the GenBank (gbk) file format but not SBOL for the description of annotated DNA products. (The concept of standardizing process description realized in QUEEN can theoretically be implemented to any semantic to describe DNA sequences.) (1) We first determined that the gbk file format is the foremost format that QUEEN needed to adopt for the life science community that already widely uses gbk files to share DNA maps (lines 282–285). (2) Second, while SBOL has been proposed to enable organized annotation of structural and functional aspects of DNA sequences in contrast to the gbk file format, we wanted to demonstrate the concept of QUEEN in the more unorganized semantic system of gbk and showcase its versatility in file format (lines 285–288).

We did not know the Process Activity Modeling Language (PAML), thank you very much. We strongly agree that an effort like PAML is extremely important to standardize experimental protocols together with the effort of laboratory automation. As we previously wrote a letter sharing the same vision to *Nature Biotechnology* (PMID: 28398329), we strongly resonate with the concept of PAML. We have now elaborated the discussion with PAML in the revised manuscript (lines 324–344).

We would sincerely appreciate it if the reviewer could kindly examine the work once again.

Reviewers' Comments:

Reviewer #1:

Remarks to the Author:

I think the authors have done an excellent job responding to my and other reviewers' comments. Their responses are detailed and considerate; the changes introduced throughout the paper and various supplementary materials have made this, IMHO, an outstanding paper.

Just two small matters:

* The authors asked for my opinion about a potential better title for the paper. I believe "A framework to efficiently describe and share reproducible DNA construction protocols" is the more concise and clear title in my honest opinion.

* In response 1.5. the authors provide a URL to a Benchling public resource but I could not actually access or find that URL in the paper or in Benchling. The authors may want to double-check the URL exists and is accessible.

I thank the authors for taking the time to make these corrections and produce this work.

Reviewer #2:

Remarks to the Author:

The authors have addressed most of my comments by significantly revising and improving it. However, I was not completely convinced regarding the following point.

"While these advances may be indirectly related to the authors' work, their limited adoption by general synthetic biology community researchers might indicate or imply the limited utility of the current work. In contrast to those previous reports that either proposed the new concept (i.e., synthetic biology framework such as standardization of interchangeable parts) for the first time, or experimentally demonstrated algorithm-guided complex circuit building, the current work seems limited in technological advances or the scope. Thus, publication in this high-impact journal would be premature, although the approach and the framework look sound."

In other words, they may need to demonstrate the utility of their work by showing some experimental work in order to be considered in this high-impact journal. Otherwise, another framework or protocol would be published without widespread adoption in the future.

Response to reviewers' comments

For “A framework to efficiently describe and share reproducible DNA materials and construction protocols” (NCOMMS-21-44872B)

Revision Summary:

Thank you very much for accepting our manuscript, now entitled “A framework to efficiently describe and share reproducible DNA materials and construction protocols.” We sincerely appreciate the constructive peer-review process to improve our manuscript and the new framework QUEEN. Below, please find our point-by-point responses to the final reviewer comments.

Reviewers' comments:

Reviewer #1 (Remarks to the Author):

I think the authors have done an excellent job responding to my and other reviewers' comments. Their responses are detailed and considerate; the changes introduced throughout the paper and various supplementary materials have made this, IMHO, an outstanding paper.

Thank you very much! We sincerely appreciate all of your thorough comments.

Just two small matters:

* The authors asked for my opinion about a potential better title for the paper. I believe "A framework to efficiently describe and share reproducible DNA construction protocols" is the more concise and clear title in my honest opinion.

Thank you very much. Because the framework is also about describing and sharing DNA materials, we have changed the title to “A framework to efficiently describe and share reproducible DNA materials and construction protocols.”

* In response 1.5. the authors provide a URL to a Benchling public resource but I could not actually access or find that URL in the paper or in Benchling. The authors may want to double-check the URL exists and is accessible.

Thank you for catching this. We have confirmed that all the correct Benchling URLs are provided in the main text and Supplementary Tables. Some of the original links with line breaks were broken after converting the original MS Word files to pdf format.

I thank the authors for taking the time to make these corrections and produce this work.

Reviewer #2 (Remarks to the Author):

The authors have addressed most of my comments by significantly revising and improving it. However, I was not completely convinced regarding the following point.

"While these advances may be indirectly related to the authors' work, their limited adoption by general synthetic biology community researchers might indicate or imply the limited utility of the current work. In contrast to those previous reports that either proposed the new concept (i.e., synthetic biology framework such as standardization of interchangeable parts) for the first time, or experimentally demonstrated algorithm-guided complex circuit building, the current work seems limited in technological advances or the scope. Thus, publication in this high-impact journal would be premature, although the approach and the framework look sound."

In other words, they may need to demonstrate the utility of their work by showing some experimental work in order to be considered in this high-impact journal. Otherwise, another framework or protocol would be published without widespread adoption in the future.

We sincerely appreciated the comments by Reviewer #2 through the peer review process, especially for the sections we could clarify the significance of QUEEN in contrast to the other standardized systems established in Synthetic Biology. We agree that the new concept has not been widely accepted by the community. At the same time, any new framework needs a debut moment, and we sincerely appreciate that the editor has agreed to publish our article. This will be a great opportunity to expose the idea of QUEEN to a broad readership. We will keep putting our best effort into disseminating the concept to the community. For example, Addgene and Benchling agreed to feature blog posts on QUEEN on their websites.